# Care and support when a baby is stillborn: A systematic review and an interpretive meta-synthesis of qualitative studies in high-income countries

Margareta Persson[1], Ingegerd Hildingsson[2‡], Monica Hultcrantz[3,4¤‡], Maja Kärrman Fredriksson[3‡], Nathalie Peira[3,4¤‡], Rebecca A. Silverstein[3‡], Josefin Sveen[2,5], Carina Berterö[6]*

1 Department of Nursing, Umeå University, Umeå, Sweden, 2 Department of Women's and Children's Health, Uppsala University, Uppsala, Sweden, 3 Swedish Agency for Health Technology Assessment and Assessment of Social Services (SBU), Stockholm, Sweden, 4 Department of Learning, Informatics, Management and Ethics, Karolinska Institute, Stockholm, Sweden, 5 Centre for Crisis Psychology, University of Bergen, Bergen, Norway, 6 Division of Nursing Sciences and Reproductive Health, Department of Health, Medicine and Caring Sciences, Linköping University, Linköping, Sweden

☯ These authors contributed equally to this work.
¤ Current address: HTA Region Stockholm, Centre for Health Economics, Informatics and Health Services Research (CHIS), Stockholm Health Care Services, Sweden
‡ IH, MH, MKF, NP and RAS also contributed equally to this work.
* carina.bertero@liu.se

## Abstract

### Introduction

Approximately 2 million babies are stillborn annually worldwide, most in low- and middle-income countries. Present review studies of the parental and healthcare providers' experiences of stillbirth often include a variety of settings, which may skew the findings as the available resources can vary considerably. In high-income countries, the prevalence of stillbirth is low, and support programs are often initiated immediately when a baby with no signs of life is detected. There is limited knowledge about what matters to parents, siblings, and healthcare providers when a baby is stillborn in high-income countries.

### Objectives

This systematic review and interpretive meta-synthesis aim to identify important aspects of care and support for parents, siblings, and healthcare professionals in high-income countries from the diagnosis of stillbirth throughout the birth and postpartum period.

### Methods

A systematic review and qualitative meta-synthesis were conducted to gain a deeper and broader understanding of the available knowledge about treatment and support when stillbirth occurred. Relevant papers were identified by systematically searching international electronic databases and citation tracking. The quality of the included studies was

**Data Availability Statement:** All relevant data are within the paper and its Supporting information files.

**Funding:** The author(s) received no specific funding for this work.

**Competing interests:** The authors have declared that no competing interests exist.

assessed, and the data was interpreted and synthesised using Gadamer's hermeneutics. The review protocol, including qualitative and quantitative study approaches, was registered on PROSPERO (CRD42022306655).

## Results

Sixteen studies were identified and included in the qualitative meta-synthesis. Experiences of care and support were interpreted and identified as four fusions. First, *Personification* is of central importance and stresses the need to acknowledge the baby as a unique person. The parents became parents even though their baby was born dead: The staff should also be recognised as the individuals they are with their personal histories. Second, the personification is reinforced by a *respectful attitude* where the parents are confirmed in their grief; the baby is treated the same way a live baby would be. Healthcare professionals need enough time to process their experiences before caring for other families giving birth. Third, *Existential issues* about life and death become intensely tangible for everyone involved, and they often feel lonely and vulnerable. Healthcare professionals also reflect on the thin line between life and death and often question their performance, especially when lacking collegial and organisational support. Finally, the fusion *Stigmatisation* focused on how parents, siblings, and healthcare professionals experienced stigma expressed as a sense of loneliness, vulnerability, and being deviant and marginalised when a baby died before or during birth. GRADE CERQual ratings for the four fusions ranged from moderate to high confidence.

## Conclusions

The profound experiences synthesised in the fusions of this meta-synthesis showed the complex impacts the birth of a baby with no signs of life had on everyone involved. These fusions can be addressed and supported by applying person-centred care to all individuals involved. Hence, grief may be facilitated for parents and siblings, and healthcare professionals may be provided with good conditions in their professional practice. Furthermore, continuing education and support to healthcare professionals may facilitate them to provide compassionate care and support to affected parents and siblings. The fusions should also be considered when implementing national recommendations, guidelines, and clinical practice.

## Introduction

From a global perspective, about one stillbirth occurs every 16th second, resulting in nearly 2 million annually [1]. Most (84%) stillbirths occur in low- and middle-income countries. The World Health Organization (WHO) further states that the experience of having a stillborn baby is often overlooked in national and international documents and agendas despite the psychological and economic consequences for the affected women and their families [2, 3].

The WHO defines stillbirth as "the death of a baby after 22 gestational weeks, and death occurred before or during birth", but the WHO also recommends using the definition of stillbirth after 28 weeks of pregnancy to facilitate international comparisons. However, many high-income countries use a lower cut-off, where a baby born without signs of life after 20–22 gestational weeks is defined as stillbirth [3]. In 2021, Hug and co-authors [4] show that the

global reduction of stillbirths at least 28 weeks gestational age is still slow compared to the worldwide improvement of the survival of children younger than five years, especially in Sub-Saharan countries. For example, high-income countries show a stillbirth rate of 3.0 stillbirths per 1000 total births, compared with 7.0 in upper-middle-income countries, 17.1 in lower-middle-income countries, and 22.7 in low-income countries in 2019 [4].

Several international research findings have stressed the need to reduce the stigma of stillbirth. A scoping review including 23 studies covering countries of low, middle, and high incomes show that the shared experiences of grieving parents are that their identities are confronted, and feelings of shame, guilt, and blame are common [5]. Furthermore, the healthcare provided for the best outcomes for bereaved parents is a limited field within the research field of pregnancy and childbirth. In their review, Ellis and co-authors [6] include 52 studies of various study designs. They show that parents' experiences are often mirrored by the staff's experiences when attending to the woman and her partner during a stillbirth. What the healthcare workers do affects the parents' memories, and concurrently, the staff indicates several barriers to providing care as wanted. Both staff and affected parents highlight measures such as better training and continuity of care. At the same time, parents also wanted supportive measures and care pathways to improve the care for parents experiencing a stillbirth [6]. Similar findings are highlighted from studies from low and middle-income countries, where basic actions such as educational interventions to the public may reduce the societal stigma of stillbirth. Additionally, respectful, supportive healthcare providers and properly investigating the causes of stillbirth may improve the experiences of affected women and their families [7].

In summary, much of today's evidence of care for bereaved parents includes a variety of countries and contexts where the stillbirth prevalence may vary considerably. There is limited information about what matters to parents, siblings, and healthcare providers when a baby is born without signs of life in a high-income context with a low prevalence of stillbirths internationally. High-income countries may have the financial resources in healthcare services to provide quality care and support programs after stillbirth, influencing the experiences of parents, siblings, and healthcare professionals. This systematic review and interpretive meta-synthesis aim to identify important aspects of care and support from the time a stillbirth is diagnosed through the birth and into the postpartum period from the perspectives of parents, siblings, and healthcare professionals in high-income countries.

## Methods

This article is part of a larger health technologies assessment covering both qualitative and quantitative aspects of care and support after stillbirth conducted by the Swedish Agency for Health Technology Assessment and Assessment of Social Services (SBU). This extensive assessment was registered in 2022 in the PROSPERO database under registration number CRD42022306655 [8]. PRISMA (Preferred Reporting Items for Systematic Reviews and Meta-analyses) guidance was followed throughout this review [9]. Please see S1 Checklist for the PRISMA checklist.

This paper is a systematic review and meta-synthesis covering a common term for an interpretive integration of qualitative findings. This integration method offers more than the sum of the individual data sets because it provides an innovative interpretation of the findings [10–12]. These interpretations are conclusions from examining all the studies in a sample as a collective group and presenting overarching interpretations not found in single studies [10]. Therefore, approximately 10 to 20 studies represent an ideal meta-synthesis [13]. This meta-synthesis followed the three processes: meta-data analysis, meta-method, and meta-theory,

outlined by Paterson et al. [12]. During meta-data analysis, the researchers critically compare the descriptions of a phenomenon to reveal similarities and discrepancies among studies. In the meta-method analysis, the researchers determine the accuracy and soundness of the research methods used in each of the studies reviewed. Finally, during the meta-theory analysis, the researchers scrutinise the underlying theoretical perspectives of each study to understand how the theories directed or influenced the findings to ensure that the findings were interpreted appropriately [12].

## Selection criteria

Before deciding on the criteria for inclusion, we sought input from representatives for non-profit organisations that support and advocate for families who have experienced stillbirth, as well as clinicians actively working or conducting research in the field, to ensure that our study question would be as relevant to them as possible. The following criteria were agreed on: qualitative and mixed-method studies addressing parents', siblings or healthcare professionals' experience of supportive care and reception from the detection and diagnosis of the stillbirth to six months postpartum were included. To be included, studies should have been conducted in high-income contexts/countries with a stillbirth rate below 5 per 1000 live births, see S1 File. Another criterion was a defined gestational age of 22 to 41 weeks at stillbirth, reflecting international definitions. Studies were excluded on the following conditions: i) addressing miscarriage, foetal anomaly, and neonatal death alone; ii) other languages than English, Danish, Norwegian, or Swedish; and iii) review articles, opinion pieces, or books.

## Search strategy

An information specialist designed the search strategy, in collaboration with the review team, based on the criteria for inclusion. After that, another information specialist at SBU reviewed the search strategy using the PRESS Checklist. A comprehensive database search was performed in December 2021 and was updated in August 2022. We searched CINAHL (EBSCO), Cochrane Library (Wiley), EMBASE (Embase.com), Medline (Ovid), and PsycINFO (EBSCO). Records retrieved from multiple databases were checked for duplicates in EndNote [14]. Hand-searching related reviews and the reference lists of the included studies supplemented the database searches. In addition, a citation search was performed via Scopus (Elsevier) to identify literature that cited any of the included studies. The entire search strategy is presented as (S2 File).

## Study selection

In pairs of two, three researchers (MP, RS, and CB) independently screened titles and abstracts using Rayyan [15], an online platform for systematic review collaboration. Two researchers (MP and CB) also independently screened the full texts according to the inclusion and exclusion criteria. Discussions within the group of authors resolved disagreements. Many of the identified articles included a mix of experiences of miscarriage, stillbirth, and early neonatal death. Still, these studies were included if the findings specific to stillbirths could be identified (e.g., citations or parts of the findings explicitly addressed stillbirth experience) and if at least 50% of participants (i.e., the majority of study participants) had experienced a stillbirth.

One of the excluded studies was a patient-led co-designed focus group study focusing on identifying improvement opportunities in healthcare. This paper was retained and later used to validate our results. Fig 1 presents the PRISMA flow diagram of the procedures [13].

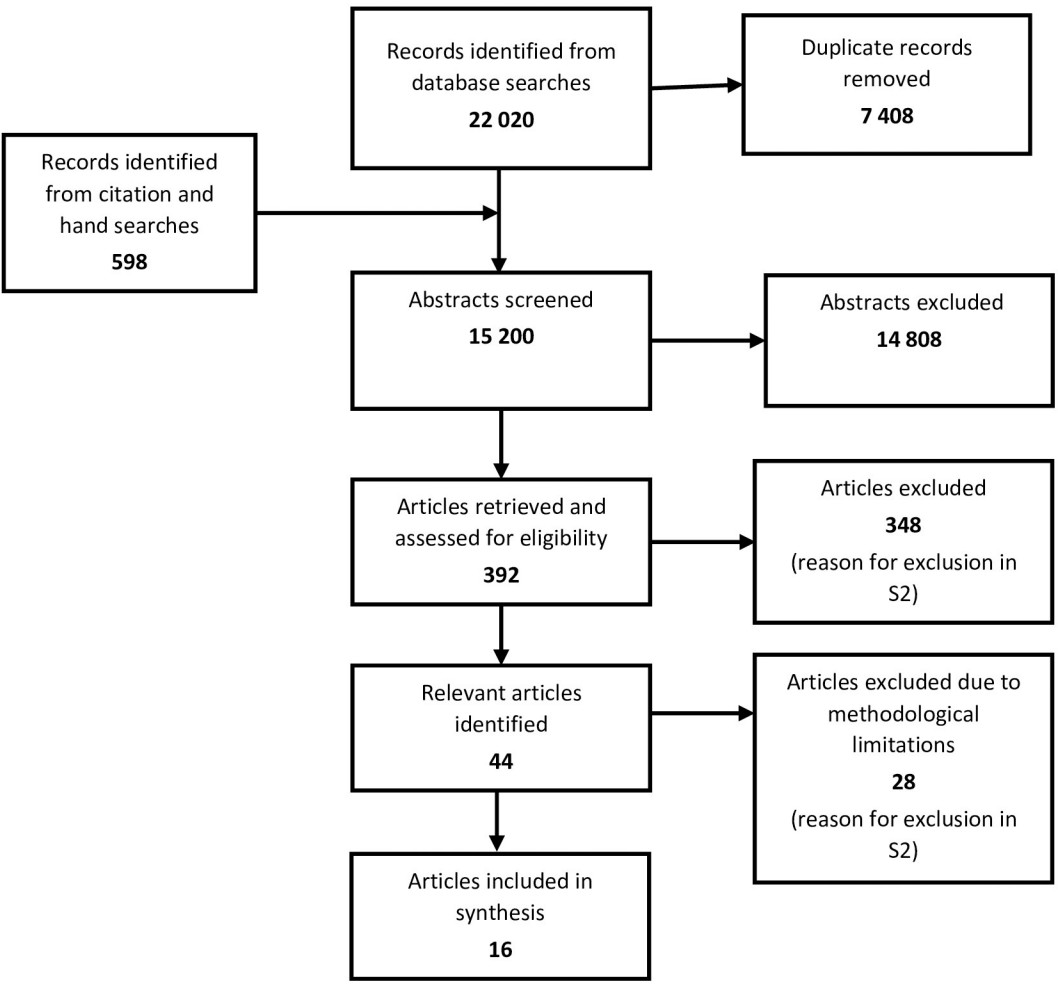

**Fig 1. Flowchart of the process.**

## Critical appraisal of full-text studies

After relevance screening, the methodological quality of the studies was appraised using the Primary Research Appraisal Tool-Qualitative (PRAT -Q) (12). The studies were systematically and independently evaluated on several aspects, such as study sample, research designs and methodology, data collection rigour and analysis, and theoretical frameworks by MP and CB. The appraisal of the studies followed a reading guide, which resulted in an individual assessment of each possible study considered for inclusion (yes/no/uncertain). All uncertainties or disagreements in the critical appraisals were discussed to reach a consensus. Only studies with no or few flaws unlikely to influence the study's trustworthiness were included. Furthermore, as all eligible studies were appraised similarly, a comparative assessment across studies was possible. This procedure allowed the development of a cross-study display, summarising key features of the studies. A final sample of 16 articles for the analysis was agreed upon for the meta-synthesis.

## Reflexive note

The researcher's active role in qualitative research is well-known since the researcher is the tool for collecting, analysing, and managing their own experiences, knowledge, and

assumptions. Therefore, it is essential to present the research team. The authors differed in disciplines and career stages and were all women. Before initiating the analysis, MP and CB discussed their pre-understanding of having researched and cared for parents of a stillborn child and their non-professional experiences of the studied phenomenon. This was done to make the researchers' potential values and perspectives visible. By applying a hermeneutical analysis approach, own experiences and understandings are an advantage or a prerequisite for the analysis [20].

## Data extraction and synthesis

Data were independently extracted from the included studies in duplicate by two authors (MP and CB) using a standardised data extraction form developed by CB. The data extraction form was piloted before use and used in other meta-thesis studies [16–19], and no changes were made to the form for this data extraction. Text related to the aim of this study was extracted, and the authors repeatedly discussed data extraction to ensure consensus and rigour.

The extracted data varied as most papers presented raw data as thematic surveys and/or direct quotations from participant interviews. We used the technique of hermeneutic appraisal to extract statements from the studies' findings to evaluate the text's horizon [20]. A horizon is a previous knowledge/prejudice found in the text or the person. We then interpreted these statements within the context of a guiding question: *Is this about care and support when babies are born dead*? We could identify possible fusions by working inductively, using our horizons to interpret the text, and then merging the horizons of the text and the author. The authors reviewed and discussed the fusions and agreed that a deepened and broadened understanding was obtained. A fusion is when both horizons from the text and the interpreter are expanding and giving a broader and deeper understanding than just answering a question.

## Confidence in synthesis findings

Once the fusions were agreed on, the level of confidence in each fusion was assessed using the Grading of Recommendations Assessment, Development, and Evaluation–Confidence in Evidence from Reviews of Qualitative Research (GRADE-CERQual) [21]. GRADE-CERQual comprises four domains that assess uncertainties in the data and include: methodological limitations, relevance, adequacy of data, and coherence. This assessment aims to evaluate and describe through a transparent procedure how much confidence there is in the findings. Therefore, an overall confidence rating of "high," "moderate," "low," or "very low" was assigned to each fusion, considering each of the four GRADE CERQual components.

All fusions except Stigmatisation were judged to have a high confidence level (Table 1). This is because they were based on rich material where the context was relevant, the data was rich, had high coherence, and only had minor methodological limitations. On the other hand, stigmatisation was considered moderate confidence as the data was not clearly described as stigmatisation in a few articles. Still, several aspects constituting the concept of stigmatisation were mentioned in these data, supporting the fusion to be labelled "stigmatisation".

## Results

### Included studies

In sum, 22 020 records were identified based on the search strategy, of which 7408 were semi-automatically removed using EndNote. With the addition of 598 abstracts retrieved from citation and hand searches, 15 200 unique abstracts were screened for relevance. Three hundred ninety-two articles were retrieved and assessed for eligibility in full text. Forty-four scientific

**Table 1. Summary of the findings and its confidence.**

| Fusion | Content, articles contributing to the fusions and number of participants | Confidence in the findings according to CERQual |
|---|---|---|
| Personification | Personification means seeing the individuals as the unique individuals they are. The child is seen as a child, a person whether alive or dead. The parents/family should be seen as the people they are and be seen as the grieving parents and relatives they are. Rituals and ceremonies are made to honor the dead child and confirm its existence, confirm parenthood and create memories for the future of the dead person. The healthcare professionals are also seen as the people they are with feelings, reactions and experiences as well as needs that come with their profession. [22–37] n = 421 | High confidence. Minor methodological limitations High coherence, there is a clear fit between the data (16 studies) and the fusion. Adequacy of data, there is rich and sufficient data to support the fusion. Relevance, the context is consistent with the research question. |
| Respectful attitude | Respectful attitude to parents and the dead child means that the parents are treated like the parents they are. Even after the child has been still born, respectful attitude means that the child is treated by the health care professionals in the same way as a living child. Respectful attitude also means that the parents are confirmed as grieving parents. It is important that the staff are also treated with respect when caring for parents and stillborn children, they need time to process their experiences before caring for other birthing women and families. [23–37] n = 413 | High confidence. Minor methodological limitations High coherence, there is a clear fit between the data (15 studies) and the fusion The missing study is the sibling study that focused on the siblings' grief processing. Adequacy of data, there is rich and sufficient data to support the fusion. Relevance, the context is consistent with the research question. |
| Existential issues | Existential questions are characterized by a storm of emotions in which parents are thrown between hope and despair, life and death. Everything is chaos in their world, what was supposed to be a joyful event has been replaced by chaos and death. Everything is uncertain; what happens, why it happens and the meaning of life is questioned. Even the siblings of the stillborn child experience the chaos and uncertainty, that life and death become tangible and raise many emotions and questions. As a healthcare professional, you become extremely aware of the fragile dividing line between life and death, which makes them value their own life, but they can also be so emotionally affected that they question their own ability and place in life. [22–37] n = 421 | High confidence. Minor methodological limitations High coherence, there is a clear fit between the data (16 studies) and the fusion. Adequacy of data, there is rich and sufficient data to support the fusion. Relevance, the context is consistent with the research question. |
| Stigmatization | Stigmatization in this situation means that the people experience loneliness and vulnerability in their situation. They have become parents, but their parenting is invisible, they leave the hospital with empty arms. Siblings experience a similar stigma, as they are alone in their grief processing and feel deviant and misunderstood by their peers. Caregivers may also experience loneliness and vulnerability in the care of parents who have a stillborn child, as collegial and organizational support is lacking. Furthermore, feelings of guilt and shame occur in everyone involved. Society's incomprehension of parental loss and grief can further contribute to the stigmatization of parents when they are met with incomprehension or marginalization of the loss among relatives, friends, health care professionals or their surroundings. [22–30] [32–36] n = 389 | Moderate confidence. Minor methodological limitations High coherence, there is a clear fit between the data (14 studies) and the fusion. However, deductions are made when the data does not so clearly express stigma, but various aspects that are included in the definition of stigmatization was found. Adequacy of data, there is rich and sufficient data to support the fusion. Relevance, the context is consistent with the research question. |

articles were considered relevant to the study question. However, 28 of those were excluded due to methodological limitations. Please see S3 File for a list of excluded articles. Sixteen articles published between 2004 and 2020 were included in the data analysis [references 22–37].

## Characteristics of included studies

Table 2 shows the primary key features of included articles. Of the 16 included articles, eleven dealt with the research question from a parental perspective [23–26, 29–30, 33–37], one study had a sibling perspective [22], and four studies covered the perspectives of healthcare professionals [27–30, 31, 32]. The studies included 380 parents, preferably mothers [23–26, 29, 30, 33–37], thirteen siblings aged 13 to 17 years [22], and 51 healthcare workers of various

**Table 2. Primary key features of included qualitative studies.**

| | |
|---|---|
| Author<br>Year<br>Country<br>Ref # | **Avelin et al**.<br>2014<br>Sweden<br>[22] |
| Aim of study | to describe adolescents' experiences of being siblings to a stillborn half-sibling |
| Underpinning theory | Grief, Bereavement |
| Setting | the Swedish National Infant Foundation |
| Participants | 13 half-siblings (11 girls, 2 boys) between 13 and 17 years (median = 14 years). Between 11 and 16 years at the time of stillbirth |
| Sampling method<br>Inclusion criteria | Self-recruited and snowball<br>Being a sibling or half-sibling to a stillborn |
| Data collection method | Face-to-face interviews at a setting chosen by the participating adolescents (home, school)<br>Each interview lasted for 30 to 90 minutes |
| Interviewer | Not described |
| Analysis methods | Content analysis (Elo & Kynas) was used to analyze data. The research team was involved in the analysis process and discussed and validated the emergent categories |
| Analysts | Nurse, Psycholgist/psychotherapeutist, midwife, midwife |
| Measures to support trustworthiness | Audit trail |
| Comments | Methodological aspects mostly fulfilled |
| Primary Research Appraisal Tool-Q (PRAT-Q) | A well-conducted qualitative study following the method stated, findings clearly presented |
| Author<br>Year<br>Country<br>Ref # | **Cacciatore et al**.<br>2013<br>Sweden<br>[23] |
| Aim of study<br>Underpinning theory | To evaluate fathers' experiences of stillbirth and psychosocial care<br>Grief |
| Setting<br>Participants | Swedish National Infant Foundation<br>131 fathers whereof most had lost their child 2006–2010. Remaining 2000–2005. |
| Sampling method<br>Inclusion criteria | Self-recruited<br>Fathers who had experienced the death of a baby to stillbirth after the 22nd week of gestation |
| Data collection method<br>Interviewer | Questionnaires, two open-ended questions. They were encouraged to write fluently about their experiences with healthcare providers in two open-ended questions<br>None |
| Analysis methods<br>Analysts | Content analysis- inductive manifest content analysis (Elo & Kyngas)<br>Not described |
| Measures to support trustworthiness | Statements were discussed by two of the authors to ensure consistency |
| Comments<br>Primary Research Appraisal Tool-Q (PRAT-Q) | Methodological aspects not fulfilled<br>A qualitative study almost following the method stated, findings clearly presented. |
| Author<br>Year<br>Country<br>Ref # | **Camacho Ávila et al**.<br>2020<br>Spain<br>[24] |
| Aim of study<br>Underpinning theory | To describe and understand the experiences of parents in relation to professional and social support following stillbirth and neonatal death<br>Grief (Worden's model) |
| Setting<br>Participants | 2 hospitals located in the southeast of Spain<br>Twenty-one parents (13 mothers and 8 fathers) from 6 families Median age 35,6 |

*(Continued)*

**Table 2.** (Continued)

| | |
|---|---|
| **Sampling method**<br>**Inclusion criteria** | Convenience.<br>parents who had suffered a stillbirth or neonatal death at least 2 years before the interview. |
| **Data collection method**<br>**Interviewer** | Contacted by first author. Face-to face interviews. The interviews had an average duration of 50 minutes<br>The first Author, PhD student? |
| **Analysis methods**<br>**Analysts** | Gadamer's Hermeneutics<br>3 researchers (Obstetricians, Midwifes, Health care worker?) |
| **Measures to support trustworthiness** | Data coding was performed individually by 3 researchers, comparing their interpretations, reach consensus. Pre-understanding, rigor, and researcher triangulation |
| **Comments**<br>**Primary Research Appraisal Tool-Q (PRAT-Q)** | Methodological aspects mostly fulfilled<br>A qualitative study almost following the method stated, findings clearly presented. |
| **Author**<br>**Year**<br>**Country**<br>**Ref #** | **Downe et al.**<br>2013<br>UK<br>[25] |
| **Aim of study**<br>**Underpinning theory** | To obtain the views of bereaved parents about their interactions with healthcare staff when their baby died just before or during labour.<br>Grief |
| **Setting**<br>**Participants** | Every National Health Service (NHS) region in the UK<br>25 participants (22 families) Mothers' age at stillbirth ranged from 18–44 |
| **Sampling method**<br>**Inclusion criteria** | purposive maximum variation sampling, from a previous survey<br>had experienced the intrauterine death or stillbirth of a baby at 24–42 weeks' gestation, between 2000 and 2010 |
| **Data collection method**<br>**Interviewer** | Qualitative in-depth interview, either face-to-face or on the telephone. Interviews lasted between 42 min and 1 h and 59 min.<br>Not described |
| **Analysis methods**<br>**Analysts** | Constant comparative technique from grounded theory (no reference). To maximize rigor, three authors read and re-read the interview transcripts individually, and then agreement was reached on<br>Midwife/researcher, social worker, medical sociologist |
| **Measures to support trustworthiness** | Reaching consensus. |
| **Comments**<br>**Primary Research Appraisal Tool-Q (PRAT-Q)** | Hard to follow the audit trail<br>A qualitative study partly following the method stated, findings presented with many quotations. |
| **Author**<br>**Year**<br>**Country**<br>**Ref #** | **Farrales et al.**<br>2020<br>Canada<br>[26] |
| **Aim of study**<br>**Underpinning theory** | To explore the experiences of grieving parents during their interaction with health care providers during and after the stillbirth of a baby.<br>participatory research |
| **Setting**<br>**Participants** | a two-day workshop on the topic of grief after stillbirth<br>Twenty-seven parents participated, comprising 12 fathers and 15 mothers with a mean age of 39 |
| **Sampling method**<br>**Inclusion criteria** | recruited from a cohort of bereaved parents<br>bereaved parents, 19 years of age or older, who experienced the stillbirth of a baby. |
| **Data collection method**<br>**Interviewe** | Four focus groups lasting 90 minutes<br>Facilitators trained in sensitive qualitative research methods who were bereaved parents or their bereaved family members and a trained research assistant (non-bereaved) was asked to attend the groups to take notes. |
| **Analysis methods**<br>**Analysts** | Geneal content analysis according to Patton<br>Psychologist, paediatrician, social worker, midwifes /research team and co-workers (bereaved parents) |

(*Continued*)

**Table 2.** (Continued)

| | |
|---|---|
| **Measures to support trustworthiness** | Co-investigators shared emergent themes with bereaved parents and HCPs in various community settings |
| **Comments**<br>**Primary Research Appraisal Tool-Q (PRAT-Q)** | Methodological aspects mostly fulfilled<br>A qualitative study following the method stated, findings clearly presented |
| **Author**<br>**Year**<br>**Country**<br>**Ref #** | **Fernandez-Alcantara et al.**<br>2020<br>Spain<br>[27] |
| **Aim of study**<br>**Underpinning theory** | To identify and examine the subjective experiences and practices of experienced professionals attending to perinatal loss in the hospital context in Spain<br> Not described |
| **Setting**<br>**Participants** | Three public hospitals in the province of Granada (Spain)<br> 16 participants: 4nurses, 2neonatologists, 1psychologist, 4 midwives, 4 nursing assistants and 1 funeral home staff member, mainly women (87.50%) mean age of 52 years (SD = 13.21) Range 33–64. Mix of type of losses participants have experienced |
| **Sampling method**<br>**Inclusion criteria** | Intentional or discriminant sampling based on maximum variation<br> being a professional in a discipline (health care or other) regularly involved in intervening in cases of perinatal loss and (ii) having at least 5 years of professional experience in attending to perinatal losses |
| **Data collection method**<br>**Interviewer** | Semi-structured interview conducted in the workplace of each participant by one researcher.<br>The mean duration of the interviews was 51 minutes, with a range of 35–88 minutes.<br> First author, psychologist |
| **Analysis methods**<br>**Analysts** | Thematic Analysis (Braun and Clarke)<br> Psychologist, psychologist |
| **Measures to support trustworthiness** | Consensus, rigor, triangulation |
| **Comments**<br>**Primary Research Appraisal Tool-Q (PRAT-Q)** | Methodological aspects mostly fulfilled<br>A qualitative study almost following the method stated, findings clearly presented. |
| **Author**<br>**Year**<br>**Country**<br>**Ref #** | **Jonas-Simpson et al.**<br>2010<br>Canada<br>[28] |
| **Aim of study**<br>**Underpinning theory** | What is the experience of caring for families whose babies were born still or died shortly after birth for obstetrical nurses?<br>The human-becoming theory, Parse, |
| **Setting**<br>**Participants** | A tertiary care urban teaching hospital where 80 registered nurses provide care to families<br>9 nurses, ranged in age from 42 to 58 years and in years of nursing experience from 13 to 24 years |
| **Sampling method**<br>**Inclusion criteria** | Flyers- Self-recruited<br>Female nurses who had cared for families who experienced perinatal loss |
| **Data collection method**<br>**Interviewer** | Interviews and the interviews lasted between 30 and 90 minutes.<br>Not described |
| **Analysis methods**<br>**Analysts** | The data analysis-synthesis process outlined in Parse<br>Nurse/researcher, Nurse, Nurse/researcher, Nurse |
| **Measures to support trustworthiness** | Team members identified and separated major ideas, reaching consensus, member checking and external auditor with specific knowledge |
| **Comments**<br>**Primary Research Appraisal Tool-Q (PRAT-Q)** | Methodological aspects mostly fulfilled<br>A qualitative study following the method stated, findings clearly presented. |
| **Author**<br>**Year**<br>**Country**<br>**Ref #** | **Lindgren et al.**<br>2014<br>Sweden<br>[29] |

(*Continued*)

**Table 2.** (Continued)

| | |
|---|---|
| **Aim of study**<br>**Underpinning theory** | to investigate mothers' experiences of saying farewell to the baby when leaving the hospital<br>Not described |
| **Setting**<br>**Participants** | The Swedish National Infant Foundation<br>23 mothers, aged ranged from 22 to 41 years |
| **Sampling method**<br>**Inclusion criteria** | Self-recruited<br>Lost a child during pregnancy |
| **Data collection method**<br>**Interviewer** | Semi-structured interviews, ranged from 52 minutes to two and a half hours<br>Not described |
| **Analysis methods**<br>**Analysts** | Qualitative Content Analysis (Lundman & Hallgren-Graneheim). The text was continuously discussed within the research team and categories were identified after consensus had been achieved. Very short description of analysis.<br>Midwife, midwife/researcher, midwife/researcher |
| **Measures to support trustworthiness** | Missing information- no audit trail |
| **Comments**<br>**Primary Research Appraisal Tool-Q (PRAT-Q)** | Methodological aspects not fulfilled<br>A qualitative study following the method stated, findings clearly presented |
| **Author**<br>**Year**<br>**Country**<br>**Ref #** | **Malm et al**.<br>2011<br>Sweden<br>[30] |
| **Aim of study**<br>**Underpinning theory** | To investigate the mothers' experiences of the time from the diagnosis of the death of their unborn baby until induction of labour<br>Not described |
| **Setting**<br>**Participants** | Swedish National Infant Foundation<br>21 mothers, age at the birth, ranged 22–41 years, babies died in utero. |
| **Sampling method**<br>**Inclusion criteria** | Self-recruited<br>Mothers who had the experience of a time gap between being informed of the diagnosis and the induction of labour |
| **Data collection method**<br>**Interviewer** | In-depth interviews, lasted between fifty minutes and two and a half hours<br>Not described |
| **Analysis methods**<br>**Analysts** | Content analysis (Graneheim & Lundman). The researchers discussed the codes, and diverging codes were re-evaluated and categories evaluated until consensus was reached<br>Midwife, midwife/researcher, midwife/researcher, midwife/researcher |
| **Measures to support trustworthiness** | Missing information |
| **Comments**<br>**Primary Research Appraisal Tool-Q (PRAT-Q)** | Methodological aspects not fulfilled<br>A qualitative study following the method stated, findings clearly presented |
| **Author**<br>**Year**<br>**Country**<br>**Ref #** | **Martínez-Serrano et al**.<br>2018<br>Spain<br>[31] |
| **Aim of study**<br>**Underpinning theory** | To explore the experiences of midwives regarding the attention given during labour in late foetal death.<br>Not described |
| **Setting**<br>**Participants** | 10 public hospitals and 1 primary health centre in Madrid, Spain<br>18 midwifes (15 female and 3 male), age range 31–54 and mean 41.27 |
| **Sampling method**<br>**Inclusion criteria** | Purposive<br>Having experience in attending cases of late foetal death |
| **Data collection method**<br>**Interviewer** | Focus groups interviews (3 groups), with a mean duration of 90 minutes, in a room at the College of Nursing<br>First author, experienced midwife, PhD candidate |

*(Continued)*

**Table 2.** (Continued)

| | |
|---|---|
| **Analysis methods**<br>**Analysts** | hermeneutic-interpretative phenomenological approach (van Manen)<br>Not described- but the final analysis included 4 of the participants |
| **Measures to support trustworthiness** | Trustworthiness was ensured through different approaches based on the framework of Lincoln and Guba, credibility, triangulation, reflexivity |
| **Comments**<br>**Primary Research Appraisal Tool-Q (PRAT-Q)** | Methodological aspects mostly fulfilled<br>A qualitative study following the method stated, findings clearly presented. |
| **Author**<br>**Year**<br>**Country**<br>**Ref #** | **Nuzum et al**.<br>2018<br>Ireland<br>[33] |
| **Aim of study**<br>**Underpinning theory** | To explore the lived experiences and personal impact of stillbirth on bereaved parents<br>The lived experience |
| **Setting**<br>**Participants** | An Irish tertiary maternity hospital<br>12 parents (12 mothers and 5 fathers). 50%-limiting diagnosis pre-birth. Remaining 50% unexpected stillbirth |
| **Sampling method**<br>**Inclusion criteria** | Purposive<br>The participants had been cared for at the study hospital, were not currently pregnant |
| **Data collection method**<br>**Interviewer** | Semi-structured in-depth Interviews lasted between 31 and 104 minutes, in a private environment without interruption at a location and time of the participants' choosing (home, or the hospital)<br>Not described |
| **Analysis methods**<br>**Analysts** | Interpretative Phenomenological Analysis. (IPA)<br>Healthcare chaplain, social scientist, consultant obstetrician (same as no 132) |
| **Measures to support trustworthiness** | Analysis by the research team- reaching consensus, Audit trail |
| **Comments**<br>**Primary Research Appraisal Tool-Q (PRAT-Q)** | Methodological aspects mostly fulfilled<br>A qualitative study following the method stated, findings clearly presented |
| **Author**<br>**Year**<br>**Country**<br>**Ref #** | **Nuzum et al**.<br>2014<br>Ireland<br>[32] |
| **Aim of study** | To explore the personal and professional impact of stillbirth on consultant obstetrician gynecologists. |
| **Underpinning theory** | The lived experience |
| **Setting** | A tertiary university maternity hospital |
| **Participants** | 8 gynecologists (equal gender balance) |
| **Sampling method**<br>**Inclusion criteria** | Purposive<br>Being permanent consultant staff and provided care for parents following stillbirth |
| **Data collection method** | Semi-structured in-depth interviews lasted between 27 and 58 minutes, in a private office environment without interruption at the participants' place of work at a time of the participants' choosing |
| **Interviewer** | Not described |
| **Analysis methods** | Interpretative Phenomenological Analysis. (IPA) |
| **Analysts** | Healthcare chaplain, social scientist, consultant obstetrician |
| **Measures to support trustworthiness** | Analysis by the research team- reaching consensus, Audit trail and member checking |
| **Comments**<br>**Primary Research Appraisal Tool-Q (PRAT-Q)** | Methodological aspects mostly fulfilled<br>A qualitative study following the method stated, findings clearly presented |

(*Continued*)

**Table 2.** (Continued)

| | |
|---|---|
| **Author**<br>**Year**<br>**Country**<br>**Ref #** | **Ryninks et al.**<br>2014<br>UK<br>[34] |
| **Aim of study**<br>**Underpinning theory** | To investigate how mothers describe their experience of spending time with their stillborn baby and how they felt retrospectively about the decision they made to see and hold their baby or not<br>Not described |
| **Setting**<br>**Participants** | Nine National Health Service (NHS) hospitals in the UK<br>21 women with a mean age of 34.4 years (SD = 5.2) |
| **Sampling method**<br>**Inclusion criteria** | Purposive<br>Women 18 and over who had experienced a stillbirth at 24 weeks gestational age or later |
| **Data collection method**<br>**Interviewer** | In depth interviews in the participants' home, interviews lasted between 20 and 30 minutes<br>five researchers—4 authors and?, (psychologists?) |
| **Analysis methods**<br>**Analysts** | Interpretive Phenomenological Analysis (IPA), following the steps according to Smith<br>Two of the authors |
| **Measures to support trustworthiness** | Credibility checks were achieved through triangulation with senior members of the research team |
| **Comments**<br>**Primary Research Appraisal Tool-Q (PRAT-Q)** | Methodological aspects mostly fulfilled<br>A qualitative study following the method stated, findings clearly presented. |
| **Author**<br>**Year**<br>**Country**<br>**Ref #** | **Radestad et al.**<br>2014<br>Sweden<br>[35] |
| **Aim of study**<br>**Underpinning theory** | To explore mothers' experiences of the confirmation of ultrasound examination results and how they were told that their baby had died in-utero.<br>Not described |
| **Setting**<br>**Participants** | Swedish National Infant Foundation<br>26 mothers of stillborn babies, 18 interviewed 1–6 years after IUFD |
| **Sampling method**<br>**Inclusion criteria** | Self-recruited<br>The mothers should have given birth to a dead child after 28 gestational weeks |
| **Data collection method**<br>**Interviewer** | Interviews at a place chosen by the women, home, or other places. Interviews lasted between 55 and 90 minutes<br>Skilled midwife, second author |
| **Analysis methods**<br>**Analysts** | Qualitative content analysis with an inductive approach (type of QA is not described)<br>Not described |
| **Measures to support trustworthiness** | Thoroughly describing the process of analysis, especially by providing a context through the use of exact quotations |
| **Comments**<br>**Primary Research Appraisal Tool-Q (PRAT-Q)** | Hard to follow the audit trail<br>A qualitative study almost following the method stated, findings clearly presented. |
| **Author**<br>**Year**<br>**Country**<br>**Ref #** | **Saflund et al.**<br>2004<br>Sweden<br>[36] |
| **Aim of study**<br>**Underpinning theory** | To focus on the caregivers' support as revealed by the parents' experiences<br>Bereavement, grief |
| **Setting**<br>**Participants** | Two regional hospitals, Karolinska Hospital and Danderyds Hospital in Stockholm<br>24 four couples and 7 mothers representing 31 stillborn children. 16 couples and 10 mothers participated in a second interview. Total data: 57 interviews. Women participants ranged from 22 to 42 years |

(*Continued*)

**Table 2.** (Continued)

| | |
|---|---|
| Sampling method<br>Inclusion criteria | Convenience<br>Parents of stillborn children at >28 weeks' gestation |
| Data collection method<br>Interviewer | Invited to study by mail, followed by telephone call. face-to-face interviews at a setting chosen by the participants (home, mothers office). Averaged 90 minutes of the first interview, the second interview focused on questions not fully addressed at the first meeting.<br>Four assistant psychologists (none of whom was involved in the actual stillborn child management work) conducted the interviews as part of their education |
| Analysis methods<br>Analysts | Qualitative Content Analysis (QSR NUD*IST) (Berg-book on qual research methods in the social sciences)<br>Social worker, Obstetrician, Nurse/researcher |
| Measures to support trustworthiness | Establishing intercoder reliability. The three researchers discussed the categorization until total agreement was reached |
| Comments<br>Primary Research Appraisal Tool-Q (PRAT-Q) | Methodological aspects mostly fulfilled. Long time between data collection and publication (data collected in 1992).<br>A qualitative study following the method stated, findings clearly presented. |
| Author<br>Year<br>Country<br>Ref # | **Trulsson, Radestad**<br>2004<br>Norway<br>[37] |
| Aim of study | To explore why induction of delivery for most women should not be delayed more than 24 hours from the diagnosis of intrauterine death. A secondary objective was to determine how the time between diagnosis and delivery should be spent |
| Underpinning theory | Not described |
| Setting<br>Participants | Ulleval University Hospital in Oslo, Norway<br>12 women who gave birth to a dead child. No age presented |
| Sampling method<br>Inclusion criteria | Purposive<br>Infant was stillborn after gestational week 24, the woman was not an inpatient at the time she was informed that the infant was dead, able to understand and speak Norwegian |
| Data collection method | Invited to study by mail. Interviews, which averaged 90 minutes, took place 6 to 18 months after the birth |
| Interviewer | Had not been involved in caring for these women |
| Analysis methods | Phenomenology, (Dahlberg) and the interviewer read the transcripts several times to gain complete understanding of the content. Participants could read and approve the transcripts. Brief description of analysis (three sentences). |
| Analysts | Midwife |
| Measures to support trustworthiness | Not described, missing information |
| Comments | Methodological aspects not fulfilled |
| Primary Research Appraisal Tool-Q (PRAT-Q) | A qualitative study but the analysis is not clear- could be a content analysis as well: Nothing about trustworthiness of findings, findings clearly presented with many quotations |

Articles about siblings are indicated with a yellow background.

Articles about healthcare staff are indicated with a grey background.

Articles about parents are indicated with a white background.

professions [27–32], which added to a total of 421 participants. The studies used varying qualitative analysis methods such as qualitative content analysis [22, 23, 26, 29, 30, 35, 37], phenomenology [24, 31–34, 37], Grounded Theory [25], and qualitative analysis according to Parse [28] and thematic analysis [27]. The studies were conducted in Ireland [31, 32], Canada [26, 28], Norway [37], Spain [24, 29, 30], the United Kingdom [25, 34], and Sweden [22, 28, 30, 35, 36]; all developed countries with a prevalence of < 5 stillbirths in 1000 live births. The study

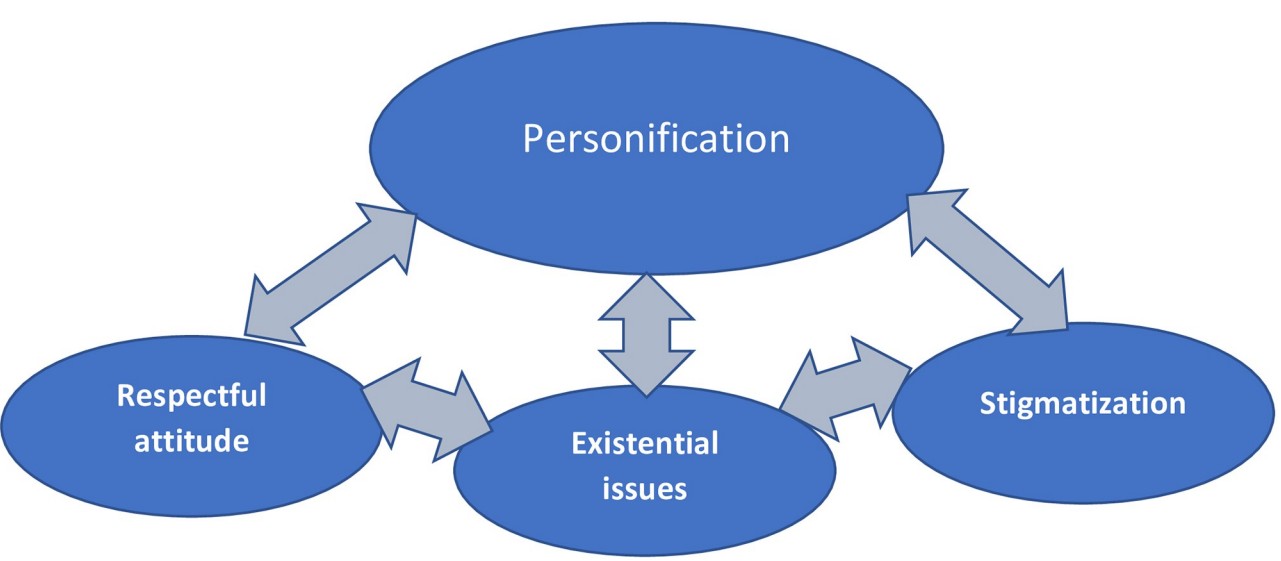

**Fig 2. Visualize the fusions and how they relate and influence each other.**

participants shared their experiences deriving from maternal health care, antenatal clinics, and other care or support bodies. In addition, the siblings' experiences included receiving information about their stillborn sibling and its subsequent consequences.

## Meta-synthesis

Our qualitative meta-synthesis resulted in four fusions: *Personification*, *Respectful attitude*, *Existential issues*, and *Stigmatisation*. The fusions are interdependent and interacting, thus bridging over to other fusions, therefore presented in the following order, reflecting a more profound and broadened understanding of what matters to parents, siblings, and healthcare professionals when a baby is born without signs of life. Please, see Fig 2 for an illustrative overview of the findings. In addition, the following presentation includes quotes from the studies that contributed to the creation of each fusion.

## Personification

The fusion "Personification" implied that the baby was considered unique, even in cases where the medically correct term would be a deceased foetus. The parents had experienced pregnancy and waited for their baby, who was now born without signs of life. Parents needed to meet their baby just as if the baby was alive to acknowledge and honour the baby. By personification, the parents could also care for their baby as parents do and have time to get to know their baby as a unique human being. Siblings and other family members could also see and connect to the baby if the parents wished. The stillborn baby was considered a unique individual and part of a family, thus treated as a family member. This personification also meant that the healthcare professionals treated the baby like any other baby, alive or dead, by, for example, addressing the baby by name, caring for it, or referring to it as a person.

*"They spoke of him as they might speak about any of the babies. They put a diaper on him (Loss in 2008)."*

[23].

Creating memories of the baby had the potential to strengthen personification. Memories of this unique individual could be created by, for example, encouraging parents to take photos of the baby on its own and with the parents and make footprints or handprints. Memories were also strengthened when parents could care for their babies for as long as they wished. Healthcare professionals could initiate and support the parents' memory creation and enable and strengthen the parental relationship with the baby. By participating in this personification, siblings could understand that a little brother or sister was born without signs of life. Furthermore, various rituals and ceremonies could also honour the baby's existence, empowering parents, siblings, relatives, and healthcare professionals. Thus, such creations of memories of the baby might facilitate the process of grief as the memories confirm that it is a unique person, a baby, who is being mourned.

*"You've got to cram a lifetime of memories into a few hours."*

*(Interview #16)* [25].

*"I kept five photos that I took and with the prints of her little feet. . . I look at the photos, I put them away, and I am left in peace. At first, I could not bear seeing, but now it is a memory that helps me in this great pain. . ."*

*(Participant 2)* [24].

Personification also included that parents were confirmed and identified as parents. Their baby was born, and the baby was dead, but they are still parents and had a personified loss and grief associated with their dead baby. Their parenthood to the dead baby was lifelong but invisible—deficiencies in recognising the unique individual and the parenthood complicated parents' and siblings' handling of the loss.

Personification also included the health care professionals' perspective. They were expected to be competent and supportive professionals providing care and support to parents in this challenging situation. At the same time, they were people with feelings, their own families, and life situations, which meant the case could become personal and affect their lives. Lack of understanding or resources in the organisation meant that healthcare professionals sometimes did not get the support they needed. This could lead to repressed emotions and distancing themselves from the care instead of having the courage and energy to stay present with the family.

*"It puts stress on you. You don't really feel like working afterwards, that's for sure because it does take a lot out of you."*

[28].

*"At first, what I was doing was hiding. Fleeing from the anxiety, and fear and ignorance. Then, I would leave to avoid everything I could, and when it was my turn, I had no choice but to stay with the woman. I had no skills or weapons to overcome my own anxiety. I gave the woman a technically correct treatment: put in her IV, give her oxytocin, with her husband or her companion, and then, leave. Because I was unable to look the woman in the face, incapable, incapable".*

*(I8—Midwife)* [27].

## Respectful attitude

The fusion 'Respectful attitude showed how crucial the caregivers' respectful attitude and care of the parents and their dead baby were for the personification and the parents' understanding

and management of their loss. But the synthesis also indicated how deficiencies in treatment and lack of information and guidance made the parents' strained situation even harder.

Respectful attitude and care of parents and the dead baby meant that the parents were treated as parents, informed, guided, and supported as in all births, regardless of whether the baby is alive or dead. A respectful attitude also included that the baby was treated and handled with the same respect and care as a live baby by the healthcare professionals. A respectful and supportive approach also covered that the woman should have the opportunity to talk about her childbirth experience after the birth and feel pride and strength that her body was able to give birth to a baby and that she was able to handle the situation. The caregivers' support, adapted information, and respectful guidance through the birth process and in the first meeting with the baby were essential to the parents. The parents were in unimaginable chaos after being informed that the baby was dead; hence decisions were difficult or impossible. They depended on being supported and guided by the healthcare professionals' empathic treatment and care. This respectful attitude, treatment and care also covered that the caregivers' compassion, respect, and empathy facilitated the meeting with the dead baby and that the parents had to say goodbye to their baby.

*"The caregivers were very competent, skilled, warm, and accommodating as they instructed us and, at the same time, kind but firm. They guided us in making the right decisions. The brief moment with the baby was very poignant. I experienced my attachment to the baby as being strengthened in that moment. It is also the only time I wept over the baby's death (father)."*

[36].

Being cared for by the same staff throughout the process meant security for the parents; they were cared for by people who knew their situation and with whom they could create a relationship.

*"All parents spoke of the relationships they had with the staff who cared for them during their pregnancy and following the birth of their baby. " I could feel the kindness off her [consultant]. I knew she really cared."*

*(2013P2)* [33].

The challenge for healthcare professionals was to provide equal care and a respectful attitude to all expectant parents, regardless of whether the baby was alive, following standard routines and treating parents and babies with warmth and respect. Thus, healthcare professionals must also be treated with respect by the organisation and colleagues when caring for parents and their stillborn babies. In addition, healthcare professionals needed time to process their experiences before caring for other birthing women and families. This respectful support was required from colleagues and the organisation.

*"Because it's so emotionally draining, sometimes you need time afterward just to regroup or talk about what you're going through."*

[28].

The disrespectful and callous treatment and care of parents or handling of the stillborn baby made the parents' meeting with the baby more difficult, gave the parents less opportunity to be and act as parents, and complicated the process of saying goodbye to the baby and mourning it.

*"The delivery was just awful from beginning to end. They almost treated me like 'The Woman With The Dead Baby'[mother's emphasis]. There was no sympathy. When I asked to see a doctor, this particular doctor came in and said, 'we're very busy.' And his exact words, I'll never forget them 'Well, with all due respect, your baby's dead already.' Which was just the most awful thing you could say."*

*(Interview #9)* [25].

## Existential issues

This fusion was characterised by a storm of emotions in which parents are thrown between hope and despair and between life and death. When the parents were informed that the baby was dead, a chaotic flood of feelings emerged—shock reactions, feelings of being thrown into an abyss, and panic wells up. At the same time, feelings of miscalculation and distrust of the information emerged and hopes for new assessments with information that the baby was alive would be presented. However, the information about the baby's death was characterised by uncertainty; the healthcare staff whispered with each other, and the parents did not seem to receive explicit information, or they were not receptive to the provided information due to the chaotic situation. Everything was chaos in the parents' world; what was to be a joyful event, a new life to be born, was now exchanged for death. Parents became mentally blocked and could not comprehend information or understand what was said. Life and justice were questioned. Why did an innocent baby die in the womb? Why was this happening to us? The questions were many but with no or few answers. This uncertainty caused frustration and anger versus healthcare professionals as the parents believed it was better to get relevant but scant information than no information as they needed something to hold on to. The information that the baby had died left the parents with enormous sadness. Concurrently, anger and resentment directed at life, the meaning of life, and the people around were experienced. The offence could be amplified when the nursing staff started to inform about funerals and rituals before the baby was born.

*"He [husband] said 'there's no heartbeat' and I said, 'we'll wait; we'll see'. I still continued in labour as if my baby could still be alive. I said, 'I'm not going to accept there's no heartbeat until I see my baby'. And then the baby came and he wasn't alive so I had no words, just no words."*

[33].

*"At the time I just wanted to die, I can't put it any other way, I just wanted to get away from the whole world, I couldn't believe all this was happening. I was in shock, I got hysterical and kept saying I had to have a cesarean and get rid of him. I didn't want to. . . but they said I had to give birth the usual way. And because of that—I didn't understand what they meant by that—I thought they were being mean to me and wanted to hurt me."*

[37].

The synthesis showed that questions of life and death became apparent to all persons in contact with the situation. First, the long-awaited baby, the new life, and the future were shattered, and then, the parents had to encounter other parents with their newborns and be noticeably reminded that their arms were empty. Even the siblings of the stillborn baby experienced chaos and uncertainty, and the thoughts of life and death became palpable. Furthermore, the

synthesis showed that the siblings' existential thoughts about the meaning of life and what mattered in life and death raised questions about their future. At the same time, the siblings might miss talking about their existential concerns as their parents were grieving.

> *"As I held Ruby, I promised her that dying wouldn't be in vain. That I would do everything I could to try and stop somebody else going through the same sort of pain."*

> *(Interview #15)*, [25].

> *"In the scope of things, death is a part of life. And it is a part of a natural occurrence in life. It just happens; it is out of our control. I think it makes you more aware of life and death and the meaning of it. And to value life more than you would otherwise"*

> *(girl aged 15)*,[22].

As birth was regarded as a predominantly joyful event—not grief and death, it contributed to a duality among the nurses and midwives, which made caring for a stillbirth difficult. Balancing being a professional healthcare worker and an empathic fellow human being could be straining. Caring for parents with a stillborn baby was regarded as a great responsibility, but it could also be a great honour to share a unique birth and have time afterwards with the family. The caregivers could express gratitude for contributing and assisting in making the birth bearable by providing empathetic care and support. Among healthcare professionals, existential issues arose, and a great awareness of the fragile dividing line between life and death was expressed. This awareness of the delicate line between life and death also aroused gratitude and joy for their family and life. However, attending to a stillbirth could be so painful and emotionally intense that healthcare professionals avoid being involved and engaged. The emotions could overwhelm them, and the whole situation lacked meaning, which affected their work and private lives and could result in questioning one's professional ability and place in life.

> *"I find it difficult to separate the job at work. . . and coming home and just switching off."*

> [32].

There were also concerns and uncertainty among healthcare professionals about what awaits parents afterwards; how should parents deal with the grief they experienced and what would wait for them after returning home from the maternity unit? Would there be any follow-up or support for the grieving parents? The nursing staff expressed that they thought about how the grieving parents could prepare to meet other pregnant women or parents with infants daily without being strongly reminded of their loss.

## Stigmatisation

This fusion presents how parents, siblings, and caregivers experienced stigma expressed as a sense of loneliness, vulnerability, and being deviant and marginalised when the baby dies before birth.

The parents felt alienated and alone, which started when they were informed about the death of their baby. This alienation continued throughout the birth process and after the baby was born. They have become parents, but their parenthood was invisible to others and society. They left the hospital with empty arms. Encountering parents with live infants reminded them of their loss and their deviation from the societal norm and other parents.

Siblings experienced a similar stigma, being alone in their grief, feeling odd and sometimes misunderstood among their peers. Additionally, they could feel lonely being part of the family as the parents were more absent due to their grief.

*"I cannot explain, but really, I thought it was terribly difficult to leave. And then we went home. And it was unbearable to leave the delivery room with only our bags and nothing else."*

*(Mother to a baby born in week 35)*,[29].

Caregivers might also experience loneliness and vulnerability when caring for parents with a stillborn baby. When collegial or organisational support was lacking, the strain was immense on each healthcare provider.

*"There was no recognition that it might be difficult. . . there was no training. . . there was no debriefing. . . and I think that's bad. You did it yourself. . . nobody cared if you got so psychiatrically disturbed you threw yourself off the roof the following week."*

[32].

Furthermore, feelings of guilt and shame appeared for the parents. These feelings originated from thinking that something they did might have contributed to the death or that they missed warning signs. It also happened that women might feel shame as their bodies failed to give birth to a live baby. Also, siblings could feel guilt when they had mixed feelings about having a new sibling, but also feelings of guilt when they wanted to live "a normal" life despite themselves and the family grieving. Healthcare professionals could feel guilty when there were suspicions that something significant in care or treatment had been missed or mistreated. These feelings were often related to thoughts about responsibility and what responsibility came with the profession.

Furthermore, society's incomprehension of parental loss further contributed to the stigmatisation of parents. Being met with belittling of one's grief from relatives, friends, healthcare professionals, or others in the society or having to hear that "it was just a foetus" or "you can have new babies" was traumatising and marginalised parental loss and grief. The extent and meaning of grief and loss were preferably shared with other parents with similar experiences.

*"Society does not understand it, it measures the pain by the size of the coffin, and you have to listen to people say things like*: *"It's okay, you'll have more children". . . it was my daughter, and she's not going to come back, and I've lost her. . . "*

*(Participant 2)*.[24].

## Validation of the findings using a patient-led co-designed focus group study

To triangulate and validate our findings, we used the results of a patient-led co-designed focus group study, where eleven participants had sessions talking about their experiences of care when their babies were born dead. Later, they reflected on the findings that would form the basis for designing practical recommendations that facilitate grief processing. After these sessions, the participants listed 15 recommendations to facilitate the parents' grieving process when affected by stillbirth [38]. These recommendations mirrored the experiences presented in our meta-synthesis to a large extent. Moreover, since the Canadian study [38] was conducted with another study focus and other study participants in a high-income context, their findings and recommendations can be suitable to compare with the results of meta-synthesis (triangulation).

Gillis and co-authors [38] summarise the parents' experiences with the words "Stillbirth, still life,"; emphasising the co-occurring parental experiences of life and death when babies are

born dead. Thus, parents need the opportunity and support to handle their experiences and grief. Our fusion "Personification" corresponded with the study findings of the parents' stories and is reflected in the recommendations provided. The study stresses the necessity that the baby is honoured as an individual and recognised as a baby. Parents also need to be supported to create memories of its existence in the brief time they share with their dead baby. By honouring the baby's existence, parents also receive a confirmation of their parenting, which coincides with the fusion of "Personification" in our meta-synthesis. Furthermore, the study [38] describes how parents are left in silence and need the empathetic treatment of healthcare professionals both in acute and long-term situations. The parents are completely unprepared for the baby's death and require time, support, and empathic guidance through all the decisions that must be taken. Suppose the health care professionals can respectfully guide and show possibilities and secure that parents are given time with their baby; in such case, the parents may not regret taking advantage of the opportunities presented to care for their baby [38]. The study also shows that the parents need to feel like a parent in connection with their baby's birth (i.e., to receive the same treatment as if the baby were alive). Simultaneously as the parents want to be treated like other expectant parents at birth, it also emphasises the need for competent and respectful nursing staff and not having to meet other parents whose babies are alive. The need for support goes beyond the acute stage, and both parents' well-being is essential to consider. Attitude is crucial for parents and their handling of grief, and they clearly remember what is said and done from the time the baby's death is confirmed and on throughout the entire period of care and follow-up. These experiences and recommendations provided by Gillis et al. [38] fit well with the fusion of "Respectful attitude" in our findings.

Life and death exist collaterally, hence raise existential issues can also be seen in the parents' narratives at Gillis et al. [38]. Parents are forced to deal with the death of their waited baby, but at the same time, the death can also raise hope about the future, but the parents have been fundamentally changed by their experience. In Gillis' study, the parents express the situation as their existence being "mid-between" during a period, which corresponds to the synthesis' interpretation of falling into chaos and needing a respectful attitude and the opportunity to express existential issues.

Our fusion "Stigmatisation" resembles the experiences described by Gillis et al. [38], who show that the parents talk about how they internalise guilt and shame. Their loss is made invisible by family and friends who cannot face the grief the parents show. They also think that they are treated differently than before. Even healthcare professionals and society contribute to reducing and making the parent's loss invisible because stillbirth has no place in society. A stillborn baby becomes the "secret" of parents, which they do not share with others [38].

## Discussion

Based on the meta-synthesis of 16 studies [ref 22–37], four fusions related to care and attitude were identified and interpreted. The four fusions identified spanned the countries, the people involved, and the period in the included papers. These vital aspects of care and attitude were characterised by the personification of the baby, its parents and siblings, and the involved healthcare professionals. A respectful attitude was essential to enable personification. Furthermore, the birth of a baby without signs of life raised several existential questions. It also implied stigmatisation and loneliness as the stillborn babies were perceived as a silenced and concealed event in society.

Other studies have stressed what mothers and fathers of stillborn babies find essential when interacting with healthcare professionals, as shown in a meta-synthesis of parents' experiences of perinatal loss [39]. Like our findings, the study indicates that perinatal loss is transformative,

whereas the parents experience multiple losses and complex emotions. The parents experience evolving relationships with their baby, which they consider a person with an identity. Healthcare professionals are an integral part of the parents' pregnancy experience and perinatal loss experience, which shows healthcare professionals' unique position to influence parents' overall experiences. The parents feel that their loss makes them traverse the social sphere in isolation and that people around ignore their pregnancy and loss [39]. Further, acknowledgement of the parenthood and parental grief, an understanding of the stillbirth's trauma, and support are emphasised even for a more extended period. Finally, the findings show how perinatal loss sets perinatal grief apart from other forms of suffering, such as grieving the loss of a parent or spouse [39], i.e., findings that align with several aspects of our fusions.

A previous review by Burden et al. [40] included 144 studies with a global approach. They show that despite country and parental gender, stillbirth experience may devastate physical and psychological health, affecting social costs and relationships. Furthermore, it influences the subsequently born children. But this review also shows that the traumatic experience may have positive effects in the form of personal growth through the development of resilience, new skills, and capacities, which resemble the existential issues described in our meta-synthesis.

Very few studies have applied the sibling's perspectives; the present studies often cover parents' need for support to siblings. Avelin et al. [41] show that it is straining for parents to try to balance their grief and simultaneously function as parents and manage everyday life. Parents try and want to include siblings in the memory creating of the dead sibling, but they need guidance and support from healthcare professionals. In a recent review [42], 25 studies were included in an integrative review of bereavement in siblings, with most focusing on loss due to the medical illness of the sibling. They conclude that the loss of a sibling is a significant childhood event with an increased risk of negative impact on physical and psychological health. In the Lancet Series of Stillbirth, Goldenberg et al. [43] mentioned bereavement support as a key action for families and communities. Still, more than a decade later, guidance and support to siblings and parents may be inadequate. Furthermore, bereavement support for children and teenagers must be age-appropriate [44].

Ellis et al. [6] show in their review that parents' and healthcare professionals' experiences in stillbirth often mirror each other. For example, parents describe how healthcare professionals hide behind tasks and routines, while staff indicates that distancing themselves from parents is their coping strategy to cope with the situation. Our findings also revealed how healthcare professionals were emotionally affected by caring for parents having a stillborn baby. Perinatal loss has a psychological impact on healthcare providers, as shown in a systematic review [45], including 20 studies focusing on the psychological impact of perinatal loss on healthcare providers. The findings show that the providers are forced to handle a variety of feelings that may, in the long term, lead to, for example, acute stress and depression. A Swedish survey [46] shows that several obstetricians and midwives report partial symptoms of post-traumatic stress syndrome after experiencing traumatic birth events, such as stillbirth, perinatal loss, etc. Staff who exhibit these symptoms often leave obstetric care and switch to work in primary care more often than their colleagues without these symptoms. Therefore, the authors conclude that targeted professional training is necessary to support healthcare professionals in managing their strains and stress. Providing empathic and quality care to parents having a stillborn child requires training and continued education. The lack of training for healthcare professionals and obstetricians has been highlighted in previous studies [47, 48]. A recently published study [49] shows that applying drama techniques in obstetrician workshop training significantly improves confidence and communication skills in breaking bad news and caring for families experiencing stillbirth. Furthermore, the workshop training also enhances the ability to

recognise own emotional reactions and support colleagues. Hence, there are reasons to emphasise that training and continued education can be essential clinical interventions to aid healthcare workers in providing quality care to parents giving birth to a stillborn baby.

Our fusion, Personification was the crucial element to perceived care and support, declares that person-centred care could be a strategy to improve care and support at stillbirths further as it focuses on the aspects of care, support, and treatment that matter most to the patient/ parents, their family, and carers. The patients, in our case, parents experiencing the unexpected loss of their baby, are not just a set of diagnoses or symptoms; instead, they are humans with emotions and social and practical needs. Person-centred care is vital for patients, but the benefits of the person-centred practice can only be fulfilled if it includes the personhood and well-being of healthcare professionals [50, 51].

## Strengths and limitations

A strength of the study is the systematic and transparent approach to the procedure and analysis, which strengthens credibility. The eligible studies were systematically and independently appraised using a Swedish PRAT-Q based on the international PRAT [12], entailing the design, selection, data collection, analysis, ethics, and theoretical frame of reference before inclusion. This appraisal also implied that only studies of no or few methodological flaws were included in the analysis. Any differences in the reviewers' assessments were discussed until consensus was established, further strengthening the result's relevance and reliability. Through interpretation, we performed a meta-synthesis to lift research results/findings to the next level and not only repeat what we already knew with some certainty [7]. The meta-synthesis used in this study was grounded in hermeneutic theories [12, 20]. In this way, we clarified the experience of the parents, siblings, and healthcare professionals when there is a stillbirth. To triangulate and validate our fusions, we used the results of a patient-led co-designed focus group study [38]. This triangulation and validation showed excellent agreement between the study findings and those in the meta-synthesis.

Some limitations in this study need addressing. Fewer studies examining the experiences of care and support from the perspectives of siblings and caregivers are available, hence in the minority of the included studies. Four of the 16 included studies deal with healthcare professionals' experiences. Only one study covers the siblings' experiences. Therefore, caregivers' and siblings' experiences may not be fully explored. Furthermore, children and young people may have different experiences and needs than their parents. Still, the siblings' experiences contributed to three of four fusions as the article contained rich information and descriptions of their experiences. Furthermore, few studies deal with the experience of health professionals in assisting parents when children are born dead. Also, in this case, the included studies contributed rich and informative data to the synthesis. Hence, the articles of siblings and health professionals significantly contributed to each fusion despite being outnumbered by parental experiences. However, the rich data from the included articles imply that the experiences of parents, siblings and caregivers may be transferable to other high-income settings. This is also supported by the triangulation and validation performed.

Another noted limitation is that the included articles have not indicated the extent to which minority groups have been represented. For example, it appears that foreign-born are included in some of the studies, but without stating whether they spoke the native language of the country or if the interview was conducted in another language or through an interpreter. Speaking the language of the country was not mentioned as an inclusion criterion for studies, which may seem relevant if interviews or surveys are conducted. When minority groups are excluded from participating in studies due to language difficulties, these groups' critical perspectives

and experiences are lost. This means that research may risk only highlighting the experiences of the majority population or among immigrant women who have been in the country long enough to master the country's language. Thus, excluding minority groups may limit the transferability of our findings to the majority populations in high-income countries.

## Implications for research and practice and directions for future research

The meta-synthesis showed that it is essential that personification works well. In the case of the stillborn baby, the baby is treated with respect as a unique individual born into a family. The siblings should be involved in care to create a relationship with the child based on their stage of development. According to the personification perspective, parents are confirmed and respected in their parenting, and other relatives and friends are welcomed to the hospital to see the baby and family. Early involvement of others may be beneficial for continued support for the family.

The meta-synthesis findings also show that healthcare professionals' need for support from colleagues and organisations must be respected. For example, they are given good conditions to focus on caring for the parents having a stillborn baby and not having to care for other patients simultaneously. Continuity with the same staff during the care period must be sought, both in the short and long term. The organisation needs to ensure that there is time and resources for reflecting conversations for the healthcare workers when they have been involved in caring for these families, in processing their feelings and getting support from the organisation in their work.

Regular training of health professionals is also vital to make them feel safe and confident in assisting the needs of parents in connection with birth. Furthermore, individual care plans can be drawn up in consultation with the parents so that the support can be designed according to the needs of the particular family regardless of, e.g., ethnic origin or belief, i.e., applying person-centred care of the family.

The existential questions of life and death arising in parents and healthcare professionals can also be essential to address and dare to respond to, as they affect the treatment of the family, parents and siblings who have lost a baby/siblings. Greater openness in society could lead to parents receiving improved societal care and support and diminish the stigma of having a stillborn baby.

Future qualitative research should endeavour to care for and support parents from different socio-economic and demographic groups to expand the current homogeneous understanding of the importance of care and support in stillbirth. These studies would include men, same-sex partners and participants from minority groups. There is also a lack of studies on care and support for siblings to a stillborn baby, as well as reflections on care and support for healthcare professionals in these situations.

With increased qualitative knowledge, the creation of relevant and adapted support measures based on theory and previous research aimed at all involved and touched when a baby is born dead. These support measures should then be systematically evaluated.

## Conclusions

The profound experiences synthesised in the fusions of this meta-synthesis showed the complex impacts the birth of a baby with no signs of life had on everyone involved. These four fusions, *Personification*, *Respectful attitude*, *Existential issues*, and *Stigmatisation* can be addressed and supported by applying person-centred care to all individuals involved. Hence, grief may be facilitated for parents and siblings, and healthcare professionals may be provided with good conditions in their professional practice. Furthermore, continuing education and

support to healthcare professionals may facilitate them to provide compassionate and person-centred care and support to affected parents and siblings. The fusions should also be considered when implementing national recommendations, guidelines, and clinical practice.

## Supporting information

**S1 Checklist. PRISMA checklist.**
(DOCX)

**S1 File. Developed countries.**
(DOCX)

**S2 File. Search strategies for qualitative articles.**
(DOCX)

**S3 File. Excluded studies.**
(DOCX)

## Acknowledgments

The authors express many thanks to Anna Attergren Granath at SBU for administrative support during the study.

## Author Contributions

**Conceptualization:** Margareta Persson, Ingegerd Hildingsson, Monica Hultcrantz, Nathalie Peira, Rebecca A. Silverstein, Josefin Sveen, Carina Berterö.

**Data curation:** Margareta Persson, Carina Berterö.

**Formal analysis:** Margareta Persson, Carina Berterö.

**Methodology:** Margareta Persson, Ingegerd Hildingsson, Monica Hultcrantz, Maja Kärrman Fredriksson, Nathalie Peira, Rebecca A. Silverstein, Josefin Sveen, Carina Berterö.

**Project administration:** Monica Hultcrantz, Nathalie Peira, Rebecca A. Silverstein.

**Resources:** Monica Hultcrantz, Maja Kärrman Fredriksson, Nathalie Peira, Rebecca A. Silverstein.

**Validation:** Margareta Persson, Carina Berterö.

**Writing – original draft:** Margareta Persson, Carina Berterö.

**Writing – review & editing:** Ingegerd Hildingsson, Monica Hultcrantz, Maja Kärrman Fredriksson, Nathalie Peira, Rebecca A. Silverstein, Josefin Sveen.

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
