## [Decision Letter · Decision Letter 0]

18 Apr 2023

PONE-D-23-03683Care and support when a baby is stillborn: A systematic review and an interpretive meta-synthesis of qualitative studiesPLOS ONE

Dear Dr. Bertero,

Thank you for submitting your manuscript to PLOS ONE. After careful consideration, we feel that it has merit but does not fully meet PLOS ONE’s publication criteria as it currently stands. Therefore, we invite you to submit a revised version of the manuscript that addresses the points raised during the review process.

We look forward to receiving your revised manuscript.

Kind regards,

Pracheth Raghuveer, MD, DNB

Academic Editor

PLOS ONE

Journal Requirements:

Reviewers' comments:

Reviewer's Responses to Questions

**Comments to the Author**

1. Is the manuscript technically sound, and do the data support the conclusions?

Reviewer #1: Yes

2. Has the statistical analysis been performed appropriately and rigorously? 

Reviewer #1: N/A

3. Have the authors made all data underlying the findings in their manuscript fully available?

Reviewer #1: Yes

4. Is the manuscript presented in an intelligible fashion and written in standard English?

Reviewer #1: Yes

5. Review Comments to the Author

Reviewer #1: Thank you for submitting this interesting qualitative review. It was an interesting read, however, I think there are some points that require exploration. I will note these in the order in which they appeared on the submitted text.

Title and Abstract:

• The title of the review does not indicate that the context refers to high income countries only nor is this apparent in the Abstract.

Abstract:

• The introduction section is a little confusing and could be more focused. The last sentence in this section, in particular, is long and confusing.

Methods:

• the opening line of the methods is not clear. What do you mean by “This article is an update of the qualitative part of a larger health technologies assessment covering both qualitative and quantitative aspects of care and support after stillbirth …”, please review.

• Line 125 on page 6, what do you mean by adjustments to the PRISMA Guidance in this context, did the review team consider also using a reporting guideline developed specifically for qualitative systematic reviews?

• Line 129 page 6 – what do you mean by a generic term in this sentence?

• While the methods advanced by Barbara Paterson are mentioned in this closing paragraph, the narrative is not explicit and I suggest that this could be reviewed and made more accessible for a reader.

Study selection:

• Why were studies in relation to miscarriage, stillbirth and neonatal death set for inclusion if 50% of the participants experienced a stillbirth? What informed this %?

Critical Appraisal:

• The manner in which the critical appraisal informed the final number of studies included in this review is not clearly articulated in this section. I suggest that you make this more explicit.

• It is also worth considering how the study selection, that occurred at this stage, impacted the characteristic of studies included in this review.

Data extraction and Analysis:

• Is the concept of synthesis missing from this heading?

• The review team refer to synthesised findings as fusions in this sections and others. Perhaps a concise explanation as to why they do this, aligned to the methodology, would be helpful for the notice researcher when reading this study.

Confidence in synthesis findings:

• The placing of this section in the manuscript could be considered so that the operationalisation of this method is included in the Methods section of the paper. And the findings or the assessment made when applying the GRADECERQual framework is noted in the findings section.

Validation of the findings using a patient-led co-designed focus group study:

• This is an interesting approach but the process is not entirely clear. Why was this study selected to inform the triangulation, how did this study relate to the eligibility criteria and screening of the review? More information in relation to the rational here would be welcome.

Discussion:

• Some repetition of narrative noted in this section, for reader comfort please review.

Strengths and Limitations:

• The opening line of this section needs to be reviewed.

• This section may need to be reviewed in the context of other feedback offered to ensure that all limitations are noted.

Summary of confidence:

• The summary of findings statements are long and contain different elements of the reported finding. It is not clear if all studies aligned in Table 2 contribute to all the elements of the finding. This needs to be reconsidered and perhaps the GRADECERQual assessment may need to be applied to a summary of findings.

6. PLOS authors have the option to publish the peer review history of their article (what does this mean?). If published, this will include your full peer review and any attached files.

Reviewer #1: No

---

## [Author Response · Author response to Decision Letter 0]

26 Apr 2023

Care and support when a baby is stillborn: A systematic review and an interpretive meta-synthesis of qualitative studies

Response to reviewer’s comments

We thank the reviewer for their comments about the manuscript and recognition that this is “an interesting paper”. Accordingly, we have revised the manuscript (shown in red colour) as requested. Point-by-point responses to the reviewer’s comments are given below.

Reviewer 1: The title of the review does not indicate that the context refers to high income countries only nor is this apparent in the Abstract.

Authors: The title and abstract have been reviewed to clarify the high-income context

Reviewer 1: The introduction section (in the abstract) is a little confusing and could be more focused. The last sentence in this section, in particular, is long and confusing.

Authors: The introduction paragraph has been revised and, hopefully, more distinct now. 

Reviewer 1; Method. the opening line of the methods is not clear. What do you mean by “This article is an update of the qualitative part of a larger health technologies assessment covering both qualitative and quantitative aspects of care and support after stillbirth …”, please review.

Authors: We used the wording update to elaborate on the qualitative part of the text in this paper. So, to not confuse the readers, we have deleted “this is an update…”

Reviewer 1: Line 125 on page 6, what do you mean by adjustments to the PRISMA Guidance in this context, did the review team consider also using a reporting guideline developed specifically for qualitative systematic reviews?

Authors: Thank you for this comment. The only adjustment done is that some items in the checklist are reported in other places than suggested in this PRISMA checklist. So, we are pleased to delete this statement. 

Reviewer 1: Line 129 page 6 – what do you mean by a generic term in this sentence?

Authors: Generic means that it is shared by or relating to a group of similar things, in this case, interpretive integration, rather than any particular method or synthesis. Synonyms are common, standard, general, etc. We have changed the word generic to common.

Reviewer 1: While the methods advanced by Barbara Paterson are mentioned in this closing paragraph, the narrative is not explicit, and I suggest that this could be reviewed and made more accessible for a reader.

Authors: Our apologies; we cannot demand that all readers should know about Paterson. We have added some text about Paterson’s three processes: meta-data analysis, meta-method, and meta-theory. Hopefully, this paragraph is now more accessible to a reader.

Reviewer 1; Study selection. Why were studies in relation to miscarriage, stillbirth and neonatal death set for inclusion if 50% of the participants experienced a stillbirth? What informed this %?

Authors: We aimed to focus on the experiences of stillbirth and no other pregnancy or infant losses for our review. Thus, we wanted to include papers where most study participants had that experience or studies that explicitly separated the findings related to stillbirth in their result presentations. We have added some additional information to this paragraph to illustrate this inclusion.

Reviewer 1; Critical appraisal. The manner in which the critical appraisal informed the final number of studies included in this review is not clearly articulated in this section. I suggest that you make this more explicit.

Authors: We have elaborated on the paragraph of the critical appraisal procedure further to show the systematical and independent manner this evaluation was performed. Hopefully, this procedure is more clearly articulated now. 

Reviewer 1: It is also worth considering how the study selection, that occurred at this stage, impacted the characteristic of studies included in this review.

Authors: Please, see the comment above. We hope that the elaborated and more detailed description has clarified any concerns. 

Reviewer 1: Data extraction and Analysis. Is the concept of synthesis missing from this heading?

Authors: Thank you for pointing this out. We changed analysis to synthesis.

Reviewer 1: The review team refer to synthesised findings as fusions in this sections and others. Perhaps a concise explanation as to why they do this, aligned to the methodology, would be helpful for the notice researcher when reading this study.

Authors: We are using Gadamer’s hermeneutics in our interpretation and synthesis. In his “method,” he discusses different horizons that merge into fusions. A theme comprises participants’ accounts characterising perceptions and/or experiences the researcher sees as relevant to the research question. A fusion is when both horizons from the text and the interpreter are expanding and giving a broader and deeper understanding than just answering a question. Additional text is added to the manuscript to explain further.

Reviewer 1: Confidence in synthesis findings: The placing of this section in the manuscript could be considered so that the operationalisation of this method is included in the Methods section of the paper. And the findings or the assessment made when applying the GRADECERQual framework is noted in the findings section.

Authors: This section is moved to the end of the methods section to show the logical flow of the analysis better. 

Reviewer 1: Validation of the findings using a patient-led co-designed focus group study:

This is an interesting approach but the process is not entirely clear. Why was this study selected to inform the triangulation, how did this study relate to the eligibility criteria and screening of the review? More information in relation to the rational here would be welcome.

Authors: We aimed to identify important aspects of care and support when babies are stillborn in high-income countries. The paper by Gillis and co-authors is performed in Canada (a high-income country with a low prevalence of stillbirth). It presents 15 recommendations to enhance bereavement care for parents based on parental experiences of stillbirth. We excluded this paper at the critical appraisal stage as it did not meet our inclusion criteria. Later we found that these recommendations mirrored our findings to a large extent. Thus, this paper which had another focus and was performed in a similar high-income country, could help discuss and validate our findings. Some short explanations are added to the manuscript. 

Reviewer 1: Discussion: • Some repetition of narrative noted in this section, for reader comfort please review.

Authors: Thank you for pointing this out. We have revised the discussion section and hope all repetition is removed. 

Reviewer 1: Strengths and Limitations: The opening line of this section needs to be reviewed.

Authors: The initial sentence in this paragraph is revised. 

Reviewer 1: This section may need to be reviewed in the context of other feedback offered to ensure that all limitations are noted.

Authors: Some revisions of the strength and limitations text sections have been made. Hopefully, the limitations of the paper are addressed now. 

Reviewer 1: Summary of confidence. The summary of findings statements are long and contain different elements of the reported finding. It is not clear if all studies aligned in Table 2 contribute to all the elements of the finding. This needs to be reconsidered and perhaps the GRADECERQual assessment may need to be applied to a summary of findings.

Authors: Thank you for your suggestion. In table 2 it is stated how many studies that contributed to the findings and those who did not are mentioned in the text

---

## [Decision Letter · Decision Letter 1]

24 Jul 2023

Care and support when a baby is stillborn: A systematic review and an interpretive meta-synthesis of qualitative studies in high-income countries

PONE-D-23-03683R1

Dear author,

We’re pleased to inform you that your manuscript has been judged scientifically suitable for publication and will be formally accepted for publication once it meets all outstanding technical requirements.

Kind regards,

Pracheth Raghuveer, MD, DNB

Academic Editor

PLOS ONE

Additional Editor Comments (optional):

Reviewers' comments:

Reviewer's Responses to Questions

**Comments to the Author**

1. If the authors have adequately addressed your comments raised in a previous round of review and you feel that this manuscript is now acceptable for publication, you may indicate that here to bypass the “Comments to the Author” section, enter your conflict of interest statement in the “Confidential to Editor” section, and submit your "Accept" recommendation.

Reviewer #1: All comments have been addressed

Reviewer #2: All comments have been addressed

2. Is the manuscript technically sound, and do the data support the conclusions?

Reviewer #1: Yes

Reviewer #2: Yes

3. Has the statistical analysis been performed appropriately and rigorously? 

Reviewer #1: N/A

Reviewer #2: Yes

4. Have the authors made all data underlying the findings in their manuscript fully available?

Reviewer #1: Yes

Reviewer #2: Yes

5. Is the manuscript presented in an intelligible fashion and written in standard English?

Reviewer #1: Yes

Reviewer #2: Yes

6. Review Comments to the Author

Reviewer #1: (No Response)

Reviewer #2: An extremely fascinating and well-organized topic for research. I want to thank you for responding to the reviewers' comments.

7. PLOS authors have the option to publish the peer review history of their article (what does this mean?). If published, this will include your full peer review and any attached files.

Reviewer #1: No

Reviewer #2: **Yes: **Mena Abdalla

---

## [Editor Report · Acceptance letter]

2 Aug 2023

PONE-D-23-03683R1 

Care and support when a baby is stillborn: A systematic review and an interpretive meta-synthesis of qualitative studies in high-income countries 

Dear Dr. Berterö:

I'm pleased to inform you that your manuscript has been deemed suitable for publication in PLOS ONE. Congratulations! Your manuscript is now with our production department. 

Kind regards, 

on behalf of

Dr. Pracheth Raghuveer 

Academic Editor

PLOS ONE